# HALENTNET: MULTIMODAL TRAJECTORY FORECASTING WITH HALLUCINATIVE INTENTS

**Deyao Zhu[1], Mohamed Zahran[12], Li Erran Li[3]\*, Mohamed Elhoseiny[1]**

[1] King Abdullah University of Science and Technology
[2] Udacity
[3] Alexa AI, Amazon and Columbia University
{deyao.zhu, mohamed.elhoseiny}@kaust.edu.sa
{mohammed.zahran}@udacity.com
{erranlli}@gmail.com

## ABSTRACT

Motion forecasting is essential for making intelligent decisions in robotic navigation. As a result, the multi-agent behavioral prediction has become a core component of modern human-robot interaction applications such as autonomous driving. Due to various intentions and interactions among agents, agent trajectories can have multiple possible futures. Hence, the motion forecasting model's ability to cover possible modes becomes essential to enable accurate prediction. Towards this goal, we introduce HalentNet to better model the future motion distribution in addition to a traditional trajectory regression learning objective by incorporating generative augmentation losses. We model intents with unsupervised discrete random variables whose training is guided by a collaboration between two key signals: A discriminative loss that encourages intents' diversity and a hallucinative loss that explores intent transitions (i.e., mixed intents) and encourages their smoothness. This regulates the neural network behavior to be more accurately predictive on uncertain scenarios due to the active yet careful exploration of possible future agent behavior. Our model's learned representation leads to better and more semantically meaningful coverage of the trajectory distribution. Our experiments show that our method can improve over the state-of-the-art trajectory forecasting benchmarks, including vehicles and pedestrians, for about 20% on average FDE and 50% on road boundary violation rate when predicting 6 seconds future. We also conducted human experiments to show that our predicted trajectories received 39.6% more votes than the runner-up approach and 32.2% more votes than our variant without hallucinative mixed intent loss.

## 1 INTRODUCTION

The ability to forecast trajectories of dynamic agents is essential for a variety of autonomous systems such as self-driving vehicles and social robots. It enables an autonomous system to foresee adverse situations and adjust motion planning accordingly to prefer better alternatives. Because agents can make different decisions at any given time, future motion distribution is inherently multi-modal. Due to incomplete coverage of different modes in real data and interacting agents' combinatorial nature, trajectory forecasting is challenging. Several existing works focus on formulating the multi-modal future prediction only from training data (e.g., (Tang & Salakhutdinov, 2019; Alahi et al., 2016; Casas et al., 2019; Deo & Trivedi, 2018; Sadeghian et al., 2018; Casas et al., 2019; Salzmann et al., 2020)). This severely limits the ability of these models to predict modes that are not covered beyond the training data distribution, and some of these learned modes could be spurious especially where the real predictive spaces are not or inadequately covered by the training data. To improve the multi-modal prediction quality, our goal is to enrich the coverage of these less explored spaces, while encouraging plausible behavior. Properly designing this exploratory learning process for motion forecasting as an implicit data augmentation approach is at the heart of this paper.

---

\*Work done prior to Amazon.

Most data augmentation methods are geometric and operate on raw data. They also have been mostly studied on discrete label spaces like classification tasks (e.g., (Zhang et al., 2017; Yun et al., 2019; Wang et al., 2019; Cubuk et al., 2019; Ho et al., 2019; Antoniou et al., 2017; Elhoseiny & Elfeki, 2019; Mikołajczyk & Grochowski, 2019; Ratner et al., 2017)). In contrast, we focus on a multi-agent future forecasting task where label space for each agent is spatial-temporal. To our knowledge, augmentation techniques are far less explored for this task.

Our work builds on recent advances in trajectory prediction problem (e.g., Tang & Salakhutdinov (2019); Salzmann et al. (2020)), that leverage discrete latent variables to represent driving behavior/intents (e.g. Turn left, speed up). Inspired by these advances, we propose HalentNet, a sequential probabilistic latent variable generative model that learns from both real and implicitly augmented multi-agent trajectory data. More concretely, we model driving intents with discrete latent variables $\mathbf{z}$. Then, our method hallucinates new intents by mixing different discrete latent variables up in the temporal dimension to generate trajectories that are realistic-looking but different from training data judged by discriminator $Dis$ to implicitly augment the behaviors/intents. The nature of our augmentation approach is different from existing methods since it operates on the latent space that represents the agent's behavior. The training of these latent variables is guided by a collaboration between discriminative and hallucinative learning signals. The discriminative loss increases the separation between intent modes; we impose this as a classification loss that recognizes the one-hot latent intents corresponding to the predicted trajectories. We call these discriminative latent intents as classified intents since they are easy to classify to an existing one-hot latent intent (i.e., low entropy). This discriminative loss expands the predictive intent space that we then encourage to explore by our hallucinated intents' loss. As we detail later, we define hallucinated intents as a mixture of the one-hot classified latent intents. We encourage the predictions of trajectories corresponding to hallcuinated intents to be hard to classify to the one-hot discrete latent intents by hallucinative loss but, in the meantime, be realistic with a real/fake loss that we impose. The classification, hallucinative, and real/fake losses are all defined on top of a Discriminator $Dis$, whose input is the predicted motion trajectories and the map information. We show that all these three components are necessary to achieve good performance, where we also ablate our design choices.

Our contributions are summarized as follows.

- We introduce a new framework that enables multi-modal trajectory forecasting to learn dynamically complementary augmented agent behaviors.
- We introduce the notion of *classified intents* and *hallucinated intents* in motion forecasting that can be captured by discrete latent variables $\mathbf{z}$. We introduce two complementary learning mechanism for each to better model latent behavior intentions and encourage the novelty of augmented agent behaviors and hence improve the generalization. The classified intents $\hat{\mathbf{z}}$ is defined not to change over time and are encouraged to be well separated from other classified intents with a classification loss. The hallucinated intents $\hat{\mathbf{z}}_{\mathbf{h}}$, on the other hand, changes over the prediction horizon and are encouraged to deviate from the classified intents as augmented agent behaviors.
- Our experiments demonstrate at most 26% better results measured by average FDE compared to other state-of-the-art methods on motion forecasting datasets, which verifies the effectiveness of our methods. We also conducted human evaluation experiments showing that our forecasted motion is considered 39% safer than the runner-up approach. Codes, pretrained models and preprocessed datasets are available at `https://github.com/Vision-CAIR/HalentNet`

## 2 RELATED WORK

**Trajectory Forecasting** Trajectory forecasting of dynamic agents has received increasing attention recently because it is a core problem to a number of applications such as autonomous driving and social robots. Human motion is inherently multi-modal, recent work (Lee et al., 2017; Cui et al., 2018; Chai et al., 2019; Rhinehart et al., 2019; Kosaraju et al., 2019; Tang & Salakhutdinov, 2019; Ridel et al., 2020; Salzmann et al., 2020; Huang et al., 2019; Mercat et al., 2019) has focused on learning the distribution from multi-agent trajectory data. (Cui et al., 2018; Chai et al., 2019; Ridel et al., 2020; Mercat et al., 2019) predicts multiple future trajectories without learning low dimensional latent agent behaviors. (Lee et al., 2017; Kosaraju et al., 2019; Rhinehart et al., 2019; Huang

et al., 2019) encodes agent behaviors in continuous low dimensional latent space while (Tang & Salakhutdinov, 2019; Salzmann et al., 2020) uses discrete latent variables. Discrete latent variables succinctly capture semantically meaningful modes such as turn left, turn right. (Tang & Salakhutdinov, 2019; Salzmann et al., 2020) learns discrete latent variables without explicit labels. Built on top of these recent work, we hallucinate possible future behaviors by changing agent intents. As the forecast horizon is a few seconds, these are highly plausible. We use a discriminator to encourage augmented trajectories to look real.

**Data augmentation** Data augmentation is a popular technique to mitigate overfitting and improve generalization in training deep networks (Shorten & Khoshgoftaar, 2019). New data is typically generated by transforming real data samples in the original input space. These transformations range from simple techniques (e.g. random flipping, mirroring for images, mixup (Zhang et al., 2017) and Cutmix (Yun et al., 2019)) to automatic data augmentation techniques (e.g. AutoAugment (Cubuk et al., 2019)) and class-identity preserving semantic data augmentation techniques (Ratner et al., 2017) (e.g. changing backgrounds of objects). Recently data augmentation via semantic transformation in deep feature space (Liu et al., 2018; Wang et al., 2019; Li et al., 2020a) has also been proposed. ISDA (Wang et al., 2019) proposes a loss function to implicitly translate training samples along with semantic directions in the feature space. For example, a certain direction corresponds to the semantic translation of "make-bespectacled." When a person's feature without glasses is translated along this direction, the new feature may correspond to the same person but with glasses. MoEx (Li et al., 2020a) proposes a new augmentation method that leverages the first and second moments extracted and re-injected by feature normalization. Specifically, it replaces the moments of the learned features of one training image by those of another and interpolates the target labels. Our data augmentation is also in the latent space, which represents agent behavior.

**Imaginative/Hallucinative models.** GANs (Goodfellow et al., 2014; Radford et al., 2015) are a powerful generative model, yet they are not explicitly trained to go beyond the training data to improve generalization. Inspired by the theory of human creativity (Martindale, 1990), recent approaches on generative models were proposed to encourage novel visual content generation in art and fashion designs. In (Elgammal et al., 2017), the authors adapted GANs to generate unconditional creative content (paintings) by encouraging the model to deviate from existing painting styles. In the fashion domain, (Sbai et al., 2018) showed that their model is capable of producing a non-existing shape like "pants to extended arm sleeves" that some designers found interesting.

The key mechanism in these methods is the addition of a deviation loss, which encourages the generator to produce novel content. More recently, (Elhoseiny & Elfeki, 2019) proposed a method for understanding unseen classes, also known as zero-shot learning (ZSL), by generating visual representations of synthesized unseen class descriptors. These visual representations are encouraged to deviate from seen classes, leading to better generalization compared to earlier generative ZSL methods. (Zhang et al., 2019) and (Li et al., 2020b) introduced methods to generate additional data based on saliency maps and adversarial learning for few-shot learning task, respectively. In the field of navigation, (Xiao et al., 2020a) and (Xiao et al., 2020b) utilized geometric information to hallucinate new navigation training data. In contrast to these earlier methods, our work has two key differences. First, our work is a sequential probabilistic generative model focusing on motion forecasting requiring time-series prediction in continuous space. Second, the deviation signal in (Elgammal et al., 2017; Sbai et al., 2018; Elhoseiny & Elfeki, 2019) is based on defining labeled discrete seen styles and seen classes, respectively. In contrast, we model the deviation from a discrete latent space guided by a deviation signal to help the model imagine driver intents without supervision signal. Similar to MoEx (Li et al., 2020a), augmented trajectories are dynamically generated during training. We believe we are the first to propose a data augmentation method in latent space for trajectory forecasting.

## 3 METHOD

**Problem Formulation** We are aiming at predicting the future trajectory $\mathbf{y}_{gt}$ of a specified agent given the input states $\mathbf{x}$, which contains the historical information like positions and heading angles of the agent itself and the surrounding agents, and a semantic map patch $\mathbf{m}$, which offers context information like drivable region, by generating a distribution $P(\mathbf{y}|\mathbf{x}, \mathbf{m})$ to model the distribution of real future trajectory $\mathbf{y}_{gt}$.

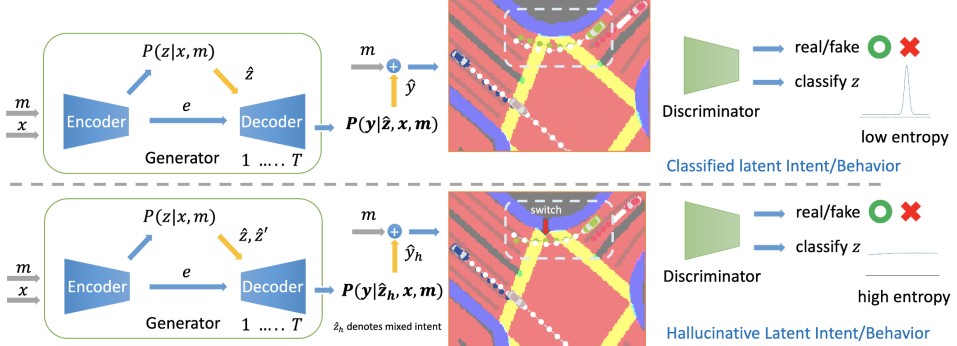

Figure 1: Overview of our architecture. The generator is trained to infer the behavior intents **z** and forecast the future trajectories. In addition to a GAN loss and a prediction loss like MLE, we propose classified latent intent behavior that classifies the latent code **ẑ** behind trajectories, and hallucinative learning that generates novel and plausible trajectories by mix two latent codes. White and color points denote the ground truth and generated trajectories, respectively.

**Our Model**   Motion forecasting in the real world is a multi-modal task. There are usually multiple possible futures given the same state. To accurately model this diversity, we define a latent code **z** to represent different intents of the predicted agent inspired by literature (e.g., (Tang & Salakhutdinov, 2019; Salzmann et al., 2020)). We denote the input state as **x**, a local map as **m**, and the corresponding ground truth future as $\mathbf{y}_{gt}$. The possible behaviors are modeled by the distribution of latent code **z** conditioned on the input state and the map $P(\mathbf{z}|\mathbf{x}, \mathbf{m})$. Then, the predicted trajectory distribution is calculated by conditioning on both input state and the latent code $P(\mathbf{y}|\mathbf{z}, \mathbf{x}, \mathbf{m})$. For motion forecasting tasks, we use maximum likelihood estimation (MLE) loss on the ground truth future as the learning objective $L = -\log P(\mathbf{y}_{gt}|\mathbf{x}, \mathbf{m})$. Note that we do not have the label for the latent code **z** in the dataset. Similar to (Tang & Salakhutdinov, 2019; Salzmann et al., 2020), we represent latent code as a discrete random variable. The learning objective can be rewritten as follows.

$$L = -\log P(\mathbf{y}_{gt}|\mathbf{x}, \mathbf{m}) = -\log \sum_i [P(\mathbf{z}_i|\mathbf{x}, \mathbf{m})P(\mathbf{y}_{gt}|\mathbf{z}_i, \mathbf{x}, \mathbf{m})] \qquad (1)$$

Hence, we obtain an unsupervised latent code $z$ that captures some uncertainty of the future without knowing its label. The distribution $P(\mathbf{z}|\mathbf{x}, \mathbf{m})$ can be modeled by any model that outputs a categorical distribution. $P(\mathbf{y}|\mathbf{z}, \mathbf{x}, \mathbf{m})$ is usually modeled by models that output multivariate Gaussian distribution. An overview of our model can be found in Fig.1. It consists of two sub-networks, a generator module and a discriminator module described in the following paragraphs.

**Generator**   The generator is the prediction model that produces the future trajectory distribution $P(\mathbf{y}|\mathbf{x}, \mathbf{m})$ given agent states **x** and a local map **m**. As the possible future is multi-modal, the output distribution should model this uncertainty. As we discussed earlier, we model distribution $P(\mathbf{z}|\mathbf{x}, \mathbf{m})$ and $P(\mathbf{y}|\mathbf{z}, \mathbf{x}, \mathbf{m})$ by neural networks. We use a discrete random variable to represent the latent code **z**. The uncertainty of the future trajectory can be factorized hierarchically into *intent uncertainty* and *control uncertainty* (Chai et al., 2019). The intent uncertainty reflects different intents or behavior modes of the agent. Furthermore, the control uncertainty covers other minor noise. As the simple Gaussian distribution $P(\mathbf{y}|\mathbf{z}, \mathbf{x}, \mathbf{m})$ is not expressive enough to model the complex uncertainty of multi-modal behaviors, this framework encourages the latent code distribution $P(\mathbf{z}|\mathbf{x}, \mathbf{m})$ to cover more the intent uncertainty. We denote the modules that generate the latent code distribution and the trajectory distribution as encoder $Enc_\theta$ and decoder $Dec_\phi$ with parameter $\theta$ and $\phi$, respectively. In addition, $Enc_\theta$ also encodes agent states **x** and the local map **m** into a feature vector **e**, which is part of the decoder's input. Note that our method does not introduce further restrictions for the model structure. Any model that fits this framework can be used as our generator like MFP(Tang & Salakhutdinov, 2019) and Trajectron++(Salzmann et al., 2020). In our experiments, we select Trajectron++ as our generator. Its original learning objective $L_{\text{traj++}}$ is shown in Appx.B. In summary,

the process of the generator can be represented by the following equations.

$$P(\mathbf{z}|\mathbf{x}, \mathbf{m}), \mathbf{e} = Enc_\theta(\mathbf{x}, \mathbf{m})$$
$$\hat{\mathbf{z}} \sim P(\mathbf{z}|\mathbf{x}, \mathbf{m}) \qquad (2)$$
$$P(\mathbf{y}|\hat{\mathbf{z}}, \mathbf{x}, \mathbf{m}) = Dec_\phi(\hat{\mathbf{z}}, \mathbf{e})$$

**Discriminator** The discriminator $Dis_\psi$ with parameters $\psi$ takes either a real trajectory $\mathbf{y_{gt}}$ or a generated one $\hat{\mathbf{y}}$ sampled from our predicted distribution together with the local map $\mathbf{m}$ as input to judge whether the trajectory is real or generated following GAN framework (Goodfellow et al., 2014). This helps the decoder inject map information into the learning signal and alleviate the violation of road boundaries in prediction. Besides, we add a classification head to the discriminator. When the input data is generated, this head needs to recognize the latent code $\hat{\mathbf{z}}$ the generator used for creating the input trajectory. In this way, the generator is forced to increase the difference among latent codes and give us more distinct and semantically meaningful driving strategies. This is further discussed in the following paragraphs. The following equation describes the function of our discriminator.

$$D(\mathbf{y}), P(\mathbf{z}|\mathbf{y}, \mathbf{m}) = Dis_\psi(\mathbf{y}, \mathbf{m}) \qquad (3)$$

Trajectory $\mathbf{y}$ is either the ground truth future $\mathbf{y}_{gt}$ or the sample from predicted trajectory distribution $\hat{\mathbf{y}}$. $D(\mathbf{y})$ is the score to indicate whether $\mathbf{y}$ is real or synthetic. $P(\mathbf{z}|\mathbf{y}, \mathbf{m})$ is the classified distribution. Our discriminator is modified from the one in DCGAN (Radford et al., 2015) by adding a fully-connected classification head at the end to classify the latent code. Trajectories $\mathbf{y}$ are transformed into the format that convolutional layers can handle via differentiable rasterizer trick (Wang et al., 2020) and stacked together with the local map $\mathbf{m}$ as the input for the discriminator as described in Appx.B.

**Learning Methods** Our architecture can be trained by a GAN learning objective, together with the original learning loss of the generator module depends on the model we choose to combine with. To learn a better behavior representation and improve the quality of predicted trajectory distribution, we introduce two new methods *Classified latent Intent Behavior* and *Hallucinative Latent Intent* for training.

**Classified latent Intent Behavior** In the real world, humans can recognize different behavior intents by looking at the trajectories. To encourage the latent code to contain more information about the intent uncertainty and less about the control uncertainty, we mimic this phenomenon and let the discriminator classify the latent code behind the generated trajectories. The classification function can be trained by a cross-entropy loss.

$$L_c = -\sum_i \hat{z}_i \log \tilde{z}_i, \quad \text{where} \quad \hat{\mathbf{z}} \sim P(\mathbf{z}|\mathbf{x}, \mathbf{m}) \qquad (4)$$

$z_i$ denotes the $i$-th dimension of the vector $\mathbf{z}$. $\hat{\mathbf{z}}$ is the latent code under the input trajectory, which is a one-hot vector sampled from the multinoulli distribution $P(\mathbf{z}|\mathbf{x}, \mathbf{m})$. $\tilde{\mathbf{z}}$ denotes the classified categorical distribution generated by our discriminator. Minimizing this loss encourages the decoder to widen the difference among predictions from different $\mathbf{z}$ to reduce the classification difficulty for the discriminator. Therefore, we reduce the overlap among output distributions from different latent codes and sharpen them to increase accuracy. Since our model is trained to classify trajectory into $\hat{\mathbf{z}}$, we name $\hat{\mathbf{z}}$ classified intent. Note that this loss is only applied for generated trajectories since we do not have the latent code for ground truth trajectory.

---

**Algorithm 1:** Training Process

Initialize $Enc_{\theta_E}, Dec_{\theta_D}, Dis_\phi$;
Initialize learning rate $\alpha, \beta$;
**while** *not converge* **do**
   // discriminator
   $\mathbf{y}_{gt}, \mathbf{x} \sim$ Dataset
   Sample normal prediction $\hat{\mathbf{y}}$
   $L_D = (1-D(\mathbf{y}_{gt}))^2 + (D(\hat{\mathbf{y}}))^2 + L_c$
   $\phi = \phi - \alpha\nabla_\phi L_D$
   // generator
   $_-, \mathbf{x} \sim$ Dataset
   Generate hallucinated trajectory $\hat{\mathbf{y}}_h$
   $\sim P(\mathbf{y}|\hat{\mathbf{z}}_\mathbf{h}, \mathbf{x}, \mathbf{m})$
   $L_{G,c} = (D(\hat{\mathbf{y}}) - 1)^2 + L_c$
   $L_{G,h} = (D(\hat{\mathbf{y}}_h) - 1)^2 + L_h$
   $\theta = \theta - \beta\nabla_\theta(\lambda L_{G,c} + (1-\lambda)L_{G,h})$
   $\mathbf{y}_{gt}, \mathbf{x} \sim$ Dataset
   $\theta = \theta - $
   $\beta\nabla_\theta(L_{\text{traj++}})$ // Trajectron++ loss
**end**

---

**Hallucinative Learning**  Latent codes $\mathbf{z}$ are trained to model intents in the training data. Each predicted trajectory $\hat{\mathbf{y}}$ is calculated from single latent code $\hat{\mathbf{z}}$ for all the prediction steps. Assume the predicted horizon is $T$, $\hat{\mathbf{y}} = [\hat{y}_1, \hat{y}_2, ..., \hat{y}_T]$, $\hat{y}_*$ at each step is generated conditioned on the same $\hat{\mathbf{z}}$. And our discriminator is trained to recognize $\hat{\mathbf{z}}$ given the synthetic trajectory $\hat{\mathbf{y}}$ by the classification loss. Besides, the MLE loss encourages synthetic trajectories to be similar to the training data. Therefore, the discriminator implicitly classifies the training data into one of the latent code $\mathbf{z}$.

We propose a novel way to utilize this property and learn beyond the training data by encouraging the model to generate trajectories from unfamiliar driving behaviors. This is done by first sampling a second different latent code $\hat{\mathbf{z}}'$ in addition to the original one $\hat{\mathbf{z}}$ and randomly selecting a time step $t_h$. The prediction until time step $t_h$ in this case ($[\hat{y}_1, ..., \hat{y}_{t_h}]$) is conditioned on the first latent code $\hat{\mathbf{z}}$ and we switch to $\hat{\mathbf{z}}'$ for the remaining steps ($[\hat{y}_{t_h+1}, ..., \hat{y}_T]$). By this way, we hallucinate a new intent by stacking 2 learned intents in the temporal dimension. We denote this mixed hallucinated intent as $\hat{\mathbf{z}}_\mathbf{h}$ and name it hallucinated intent. The predicted distribution from such a intent is denoted as $\mathrm{P}(\mathbf{y}|\hat{\mathbf{z}}_\mathbf{h}, \mathbf{x}, \mathbf{m})$. We aim to encourage the hallucinative trajectories $\hat{\mathbf{y}}_\mathbf{h}$ to be plausible but different from the training data. To achieve this, we minimize the cross entropy between the uniform distribution and our intent class distribution.

$$L_h = -\sum_i \frac{1}{N} \log \tilde{z}_i \tag{5}$$

$N$ indicates the number of latent codes. $\tilde{z}_i$ is the $i$-th dimention of the classified distribution $\tilde{\mathbf{z}}$. It encourages the hallucinative trajectory to be hard to be classified into any latent code $\mathbf{z}$, and therefore, to be different from the training data. The plausibility of the hallucinative trajectory is encouraged by the additional GAN loss. In this way, we implicitly apply data augmentation in the latent space to train a more powerful discriminator and improve the generator prediction quality. We call this method hallucinative learning inspired from literature (e.g., Hariharan & Girshick (2017)).

**Training**  We use LSGAN(Mao et al., 2017) loss with spectral normalization(Miyato et al., 2018) as our GAN learning objective. We also keep the original Trajectron++ learning loss $L_{\text{traj++}}$ to maintain the performance in case Trajectron++ is our generator. The combination of GAN learning, training of the original generator, classified latent intent behavior, and hallucinative learning is demonstrated in Alg.2 (Detailed version in Appx.F). We use a hyperparameter $\lambda$ to balance the training between classification learning and hallucinative learning for the generator by adjusting the weighting of the learning loss.

## 4    EXPERIMENTAL RESULTS

We compare the performance of our method with state-of-the-art models. To demonstrate our method's performance in complex scenarios, we focus on evaluating the nuScenes dataset (Caesar et al., 2019a) which contains about 1000 driving scenes in 2 cities (Boston and Singapore) with dense traffic. Each scene of them has annotations for pedestrians and vehicles, sampled at a rate of 2 Hz, and about 20 seconds long (40 frames). Besides, both cities include maps, which are required in our method. In addition, we also evaluate our method on widely-used pedestrian datasets ETH (Pellegrini et al., 2009) and UCY (Leal-Taixé et al., 2014).

**Evaluation Metrics**  We use average $l_2$ displacement error (ADE) and final $l_2$ displacement error (FDE) to evaluate the prediction performance. Each of them contains some sub-versions. ADE-ML/FDE-ML is the ADE/FDE calculated using the most likely predicted trajectories. In minADE-k/minFDE-k, we select k candidate trajectories for each prediction and use all candidates' minimal value as the final score. ADE-Full/FDE-Full represents the quality of output distribution. To compute ADE-Full/FDE-Full, we randomly sample 2k trajectories and calculate the average score.

**Model Setting**  Our models are trained in two different scenarios. In the first scenario, we train the model totally from scratch, and in the second one, we finetune on a pretrained generator and train the discriminator from scratch. The number of latent code $\mathbf{z}$ is set as 25 latent codes following (Salzmann et al., 2020). Our method is trained for 23 epochs with the pretrained generator and 35

Table 1: **nuScenes**: Vehicle prediction

| Model | FDE | | | |
|---|---|---|---|---|
| | 1s | 2s | 3s | 4s |
| Const. Velocity | 0.32 | 0.89 | 1.70 | 2.73 |
| S-LSTM | 0.47 | - | 1.61 | - |
| CSP | 0.46 | 2.35 | 1.50 | - |
| CAR-Net | 0.38 | - | 1.35 | - |
| SpAGNN | 0.36 | - | 1.23 | - |
| Trajectron++ | **0.07** | 0.45 | 1.14 | 2.20 |
| Ours | $0.08 \pm 0.02$ | $0.45 \pm 0.01$ | $\mathbf{1.09 \pm 0.01}$ | $\mathbf{2.03 \pm 0.02}$ |

Table 2: **nuScenes**: Pedestrian prediction

| Model | FDE | | | |
|---|---|---|---|---|
| | 1s | 2s | 3s | 4s |
| Traj++ | **0.01** | 0.17 | 0.37 | 0.62 |
| Ours | 0.02 | 0.17 | **0.35** | **0.57** |

Table 3: **nuScenes**: Detailed comparison with Trajectron++. $\Delta\%$ is relative improvement.

| Metric | Model | 1s | 2s | 3s | 4s | 5s | 6s |
|---|---|---|---|---|---|---|---|
| FDE (Full) | Traj++ | 0.16 | 0.64 | 1.52 | 2.80 | 4.53 | 6.70 |
| | Ours | **0.09** | **0.52** | **1.21** | **2.17** | **3.41** | **4.93** |
| | $\Delta\%$ | +43 | +19 | +20 | +23 | +25 | +27 |
| FDE (ML) | Traj++ | 0.06 | 0.41 | 1.06 | 2.06 | 3.46 | 5.29 |
| | Ours | **0.05** | **0.40** | **1.00** | **1.88** | **3.06** | **4.52** |
| | $\Delta\%$ | +16 | +2 | +6 | +9 | +12 | +14 |
| minFDE-10 | Traj++ | 0.05 | 0.31 | 0.65 | 1.15 | 1.75 | 2.57 |
| | Ours | **0.04** | **0.30** | 0.65 | **0.99** | **1.49** | **2.24** |
| | $\Delta\%$ | +20 | +3 | 0 | +14 | +15 | +13 |
| minADE-10 | Traj++ | **0.06** | 0.15 | 0.29 | 0.47 | 0.71 | 1.02 |
| | Ours | 0.05 | 0.15 | **0.28** | **0.42** | **0.61** | **0.89** |
| | $\Delta\%$ | -16 | 0 | +3 | +10 | +14 | +13 |
| RB. Viol. | Traj++ | **0.24%** | 0.57% | 2.55% | 7.04% | 12.95% | 19.09% |
| | Ours | 0.26% | **0.45%** | **1.30%** | **3.21%** | **6.00%** | **9.22%** |
| | $\Delta\%$ | -8 | +21 | +49 | 54 | +54 | +52 |

epochs from scratch for vehicles. The training with a pretrained model lasts about 16 hours with a single NVIDIA V100 graphic card and about 24 hours from scratch.

**Comparison Methods**    We compare our contribution to state-of-the-art methods. **S-LSTM** (Alahi et al., 2016) uses LSTM to predict trajectories and pool the hidden states among agents to model their interaction. **CSP** (Deo & Trivedi, 2018) discretizes behaviors into a fixed number of classes and predict the best possible behaviors. **CAR-Net** (Sadeghian et al., 2018) utilizes visual attention mechanism to encodes the surrounding environment and **SpAGNN** (Casas et al., 2019) detects agents first from LIDAR and semantic map. Then, a graph neural network decoder interactively predicts their trajectories. **Trajectron++** (Salzmann et al., 2020) encodes surrounding vehicles using a graph neural network model and infers the behavior intents to produce a multi-modal prediction. As Trajectron++ is the best model among these baselines, we perform an extensive comparison with it using the released pretrained model.

**nuScenes Dataset**    We run extensive experiments on the nuScenes dataset (Caesar et al., 2019b) to evaluate and analyze our trajectory forecasting performance and verify model ability to learn dynamically complementary augmented agent behaviors. In this task setting, the model forecasts 3 seconds future with maximal 4 seconds of history information during training. However, the prediction horizon for evaluation is up to 6 seconds to demonstrate our model's generalization capacity. NuScenes dataset contains many agent categories like adult pedestrian and truck. We group them into two semantic classes *vehicle* and *pedestrian*, train individual models on them and report the performance separately, following (Salzmann et al., 2020).

Our method achieves the best performance compared to other state-of-the-art approaches on the FDE with minimal 4 seconds of future information during testing; see Tab.1. Due to the instability of GAN, we remove the diverging training cases and average the numbers over 3 stable runs. Although other methods do not report values at 2s and 4s, we can see that the performance of HalentNet increases and HalentNet outperforms existing approaches as we predict more time steps in the future. The complementary learning mechanism and hallucinated intents show a noticeable improvement in vehicle trajectory prediction.

We run more experiments to examine further our method's performance and Trajectron++ (Salzmann et al., 2020) as Trajectron++ outperforms other baseline approaches. We used various metrics with the prediction horizon from 1 second to 6 seconds for all tracked objects with at least 6-second available future data. The evaluation results are demonstrated in Tab.3. Our method outperforms Trajectron++ in almost all metrics with a significant margin. Besides, the methods also generalize well when we extend the prediction horizon. We obtain about $26\%$ on average FDE over the output distribution (FDE Full) and $52\%$ for the road boundary violation improvement over the baseline model in the 6-second prediction case. Superiority is when the prediction horizon is more extended. HalentNet trajectories show more respect to road boundaries and output plausible trajectories produced from hallucinated intents that are changed over the prediction horizon and are encouraged to deviate from the classified intents. The evaluation on the pedestrian nuScenes benchmark is listed in Tab.2. We obtain a 8% improvement in the 4s prediction horizon case.

Table 4: **With Map.** We evaluate our model and Trajectron++ (both trained from scratch) on the combination of ETH and Hotel datasets with maps. Maps help to learn a better discriminator, hence increase the performance of our method on ETH and Hotel sets. Note that Trajectron++ also takes maps as input in this experiment for a fair comparison. Our method is significantly better.

| Methods | ADE ML | FDE ML | minADE-20 | minFDE-20 |
|---|---|---|---|---|
| Trajectron++ | 0.48 | 1.15 | 0.37 | 0.89 |
| Ours | **0.46** | **1.08** | **0.31** | **0.70** |

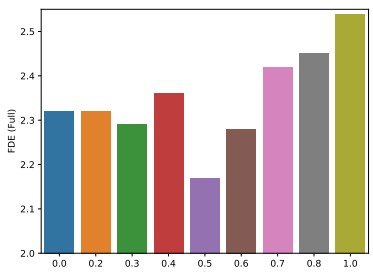

Table 5: Ablation study on **nuScenes** dataset.

| Components | | | FDE Full | | | FDE ML | | | B. Violations | | |
|---|---|---|---|---|---|---|---|---|---|---|---|
| $Dis$ | $L_c$ | $L_h$ | 3s | 4s | 5s | 3s | 4s | 5s | 3s | 4s | 5s |
| + | + | + | **1.21** | **2.17** | **3.41** | **1.00** | **1.88** | **3.06** | **1.30%** | **3.21%** | **6.00%** |
| + | - | + | 1.23 | 2.32 | 3.81 | 1.01 | 1.90 | 3.09 | 1.48% | 4.91% | 10.84% |
| + | + | - | 1.29 | 2.54 | 4.27 | 1.02 | 1.99 | 3.31 | 1.75% | 6.09% | 12.44% |
| + | - | - | 1.28 | 2.35 | 3.75 | 1.08 | 2.01 | 3.23 | 1.64% | 4.62% | 8.66% |
| - | - | - | 1.52 | 2.80 | 4.53 | 1.06 | 2.06 | 3.46 | 2.55% | 7.04% | 12.95% |

Figure 2: Balancing classified latent intent behavior and hallucinative learning by selecting a proper $\lambda = 0.5$ in Algo.2 helps to get a best FDE ( averaged over 2000 random samples)

**Pedestrian Datasets** To further demonstrate our performance, we train our model on two widely used pedestrian datasets; ETH(Pellegrini et al., 2009) and UCY(Leal-Taixé et al., 2014).

*UCY (no map).* UCY does not provide map information that is important for our method. We still test our method in this case in Tab. 10 in Appendix E. This can be viewed as a variant of our model since the map is not provided. We observe a slight improvement in the FDE results with about 7% over Trajectron++. As we show later, the improvement is more significant when map information is used that we think is available in most cases.

*ETH (with map).* We split the data by 70%, 15%, and 15% as a training set, validation set, and test set, separately. Then, we combine these two sets as one big dataset and train both our method and Trajectron++ from scratch with map information. The assessment uses an observation period of 8 timesteps (3.2s) and a projected horizon of 12 timesteps (4.8s). The results are shown in Tab.4. Our method is significantly better than Trajectron++ with an improvement of about 20% on minFDE-20.

**Ablation Study** To better demonstrate and understand each component's effect in our model, we create model variants by removing the evaluated components step by step and showing their performance. The evaluation is on the nuScenes dataset with the vehicle prediction for all tracked objects with at least 6-second available future data. The results are listed in Tab.5. $Dis$, $L_c$, and $L_h$ denote the discriminator, the classification learning, and the hallucination learning, respectively.

Compared to the variant without all the components we list, the model with the discriminator outperforms it by 15% on average FDE over the output distribution (FDE Full) and 30% for the road boundary violation. The FDE of most likely prediction is also better after 3 seconds. This indicates that the discriminator helps to improve the quality of output distribution. One of the possible reasons is the injection of the map info. Although the generator takes the local map $\mathbf{m}$ as input, we do not guarantee that the plain model will use it. As a trajectory that violates the road boundary can be easily recognized as fake data by the discriminator, the map info is injected into the GAN learning objective. Hence, optimizing this loss helps to push map information into output distribution. We observe that we can not gain additional improvement when we add the classification loss $L_c$. We think this is because $L_c$ only encourages the classified intents $\hat{\mathbf{z}}$ to be distinguishable from each other. And this property doesn't have a clear relationship to the performance. Our method benefits from the implicit behavior augmentation by the hallucinated intent. When we implicitly augment the data by hallucinative intent loss $L_h$, mixing intents during training with $\hat{\mathbf{z}}_h$ by combining classified intents, we observe a further boost in the performance. The FDE is more than 5% better than the discriminator only variant $Dis$, and the road boundary violation is about 20-30% better, showing the effectiveness of the hallucinative learning; see Table 5. Note that although $L_c$ alone does not improve, it is still important to encourage the hallucinative signal $L_h$ to be more explorative. This is since the exploration of $L_h$ depends on the classified intents' diversity that $L_c$ increases. Mathematically, the classification loss $L_c$ encourages reducing the entropy of the categorical output distribution over $\mathbf{z}$, and the hallucinative loss promotes that mixing these intents can still be plausi-

Table 6: Ablation study on different designs for hallucinative loss $L_h$ on nuScenes.

| Methods | FDE Full | | | FDE ML | | |
|---|---|---|---|---|---|---|
| | 3s | 4s | 5s | 3s | 4s | 5s |
| $L_h$ (default=entropy) | 1.21 | **2.17** | **3.41** | **1.00** | **1.88** | **3.06** |
| $L_h$ (mixup) | **1.19** | 2.20 | 3.60 | 1.02 | 1.97 | 3.60 |
| $L_h(N+1)$ | 1.35 | 2.69 | 4.58 | 1.20 | 2.63 | 4.66 |

Table 7: Human Evaluation Results

| Compared Methods | Votes for our method | Votes for compared method | Our Advantage |
|---|---|---|---|
| Variant without $L_h$ | 427 | 323 | 32.2% |
| Trajectron++ | 437 | 313 | 39.6% |

ble. In other words, $L_c$ trains the discriminator to classify trajectories and $L_h$ trains the generator to output trajectories that are hard to be classified if we use hallucinated intent. If we remove $L_c$ and keep $L_h$ only, our classifier cannot be trained, which make $L_h$ meaningless. Hence, the two losses are complementary to one another. The complementary importance of $L_c$ to $L_h$, can be explained by drop in performance when $L_c$ only is discarded (second row in Tab 5).

Fig.2 shows our exploration of how to balance the classification learning and hallucinative learning. $\lambda$ represents the importance of classification learning. When $\lambda = 1$, our method is reduced to the variant without hallucinative learning. We set the $\lambda$ in Alg.2 from 0.0 to 1.0 for training separately and plot the corresponding average FDE over the output distribution. The results suggest that properly balancing classified latent intent behavior and hallucinative learning helps improve performance.

Hallucinative loss $L_h$ defined in Eq.5 is used to encourage the classification difficulty of the hallucinated trajectory $\hat{\mathbf{y}}_h$. $L_h$ is defined as the cross entropy between the uniform distribution and the classification results in our method. Here we denote our original design choise as $L_h(default)$ In addition to this design choice, we also experiment with another 2 possibilities: $L_h(mixup)$ and $L_h(N+1)$. $L_h(mixup)$ is defined as the cross entropy between a discrete distribution that only has non-zero probabilities on the 2 latent codes (probabilities equal 50% for both) used together as the hallucinated intent $\hat{\mathbf{z}}_h$ and the classification results. For $L_h(N+1)$, we define a new class label for all the hallucinated trajectories. $L_h(N+1)$ is the cross entropy between this new class and the prediction results. Results are shown in Tab.6. Our design choice achieves the best performance, but the TwoHot variant also shows comparable results. The performance of AdditionClass is much worse compared to our design and TwoHot.

**Human Evaluation** We use Amazon Mechanical Turk to evaluate the quality of our prediction. We randomly selected 150 paired scenes, each of which is evaluated by five human subjects on MTurk who are requested to judge which model predicts better trajectory given a scene. Each scene is evaluated by 5 times. Therefore, each comparison contains 750 votes in total. Our method generates better trajectories compared to our variant without hallucinative learning measured by 32.2% more votes and Trajectron++ measured by 39.6% more votes. Results shown in Tab.7.

## 5 CONCLUSION

In this paper, we propose HalentNet, a probabilistic latent variable framework that hallucinates novel trajectories via transformations in discrete latent agent behavior space. Our method contains two complementary learning mechanisms that encourage a diverse and novel generation to regulate the neural network behavior and achieve more accurate predictions on uncertain scenarios. We show that HalentNet can significantly improve generalization for multi-modal future predictions in multi-agent settings and reduces the boundary violation metric by more than 50%.

## 6 ACKNOWLEDGEMENT

This work is funded by a KAUST BAS/1/1685-01-0. The authors wish to thank Amazon Mechanical Turkers without who helped with our human studies.

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

## A   APPENDIX

In this document, we present more explanations and details about the training and the testing results on the NuScenes dataset and the pedestrian datasets with more visualization for our method. It contains the following sections:

- Training Details
- Qualitative Results
- Pedestrian Experiments

The code and pretrained models will be released in the future (soon).

## B   TRAINING DETAILS

**Original Loss of Generator**   In this dataset, we initialize the generator of our Halent model with the pretrained Trajectron++ model (Salzmann et al., 2020). The original Trajectron++ training loss $L_{\text{traj}}$ is kept in our method.

$$L_{\text{traj++}} = -\mathbb{E}_{\hat{\mathbf{z}} \sim q(\mathbf{z}|\mathbf{x},\mathbf{m},\mathbf{y}_{gt})}[\log p(\mathbf{y}_{gt}|\mathbf{x},\mathbf{m},\hat{\mathbf{z}})] + k_1 D_{KL}(q(\mathbf{z}|\mathbf{x},\mathbf{m},\mathbf{y}_{gt})||p(\mathbf{z}|\mathbf{x},\mathbf{m})) - k_2 I_q(\mathbf{x};\mathbf{z}) \tag{6}$$

Here, $k_2$ is set to 1. Instead of directly learning the distribution of latent intents $p(\mathbf{z}|\mathbf{x},\mathbf{m})$, Trajectron++ learns $q(\mathbf{z}|\mathbf{x},\mathbf{m},\mathbf{y}_{gt})$ which additionally conditioned on ground truth trajectory during training. $p(\mathbf{z}|\mathbf{x},\mathbf{m})$ is learned by reducing the KL divergence between $q(\mathbf{z}|\mathbf{x},\mathbf{m},\mathbf{y}_{gt})$ and $p(\mathbf{z}|\mathbf{x},\mathbf{m})$. $k_1$ is gradually increase to enhance the information transfer. Note that only $p(\mathbf{z}|\mathbf{x},\mathbf{m})$ is used during testing.

**Differentiable Rasterizer**   To combine the trajectory $\mathbf{y}$ and the local map $\mathbf{m}$ into a acceptable format for the CNN-based discriminator, we use differentiable rasterizer (Wang et al., 2020) to convert $\mathbf{y}$, which can be represented by a sequence of $T$ positions $\{(x_1,y_1),(x_2,y_2),...(x_T,y_T)\}$, into $T$ 2D occupancy grids $\{\mathcal{G}_1,\mathcal{G}_2,...\mathcal{G}_T\}$. Each grid $\mathcal{G}_t$ is a tensor with the same weight and height of $\mathbf{m}$. In detail, it creates a bivariate Gaussian distribution $\mathcal{N}(\mu_t,\Sigma_t)$ for every time step $t$, where $\mu_t = f_a(x_t,y_t)$, $\Sigma = diag(\sigma^2,\sigma^2)$. $\sigma$ is a hyperparameter. The value for cell $(i,j)$ of $\mathcal{G}_t$ is the scaled probability density at location $(i,j)$ in the map coordinate system

$$\mathcal{G}_t[i,j] = k \cdot \frac{\mathcal{N}((i,j)|\mu_t,\Sigma_t)}{\mathcal{N}(\mu_t|\mu_t,\Sigma_t)} \tag{7}$$

Here, we normalize the occupancy grids so the maximal amplitude equal to $k$. By this way, we obtain 2D trajectory grids $\{\mathcal{G}_1,\mathcal{G}_2,...\mathcal{G}_T\}$, which can be processed by CNN and are differentiable w.r.t the original trajectory. In our experiments, we set $k = 9$ and $\sigma = 5$ based on the hyperparameter search on the validation set.

**Training**   We set $\lambda = 0.5$ to balance the classified latent intent behavior and hallucinative learning. The model is trained by Adam optimizer (Kingma & Ba, 2014). The pretrained model is trained by 12 epochs. We continued the training for another 23 epochs with our method and kept the original learning rate for our generator. The learning rate of the discriminator is lower compared to the generator to avoid a large gradient at the beginning of training.

## C   ADDITIONAL RESULTS ON NUSCENES

Here we report the ADE scores of our method compared to Trajectron++ and the variants of our method in Tab.8 as a supplementary to Tab.1 and Tab.5. The evaluation is on the nuScenes dataset with the vehicle prediction for all tracked objectswith at least 4-second available future data. Compared to Trajectron++ (the last row), we obtain a 25cm improvement in the 4s case measured by ADE-Full, which is about 21% better.

Table 8: ADE scores on **nuScenes** dataset.

| Components | | | ADE Full | | ADE ML | |
|:---:|:---:|:---:|:---:|:---:|:---:|:---:|
| $Dis$ | $L_c$ | $L_h$ | 3s | 4s | 3s | 4s |
| + | + | + | **0.53** | **0.91** | **0.43** | **0.76** |
| + | - | + | **0.53** | 0.94 | 0.44 | 0.77 |
| + | + | - | 0.55 | 1.00 | 0.44 | 0.79 |
| + | - | - | 0.56 | 0.97 | 0.47 | 0.83 |
| - | - | - | 0.67 | 1.16 | 0.45 | 0.82 |

Table 9: Results for the variant without discriminator map input on nuScenes.

| Methods | FDE Full | | | FDE ML | | |
|:---:|:---:|:---:|:---:|:---:|:---:|:---:|
| | 3s | 4s | 5s | 3s | 4s | 5s |
| Ours | 1.21 | 2.17 | 3.41 | 1.00 | 1.88 | 3.06 |
| Ours without discriminator map input | 1.31 | 2.38 | 3.75 | 1.07 | 1.99 | 3.18 |
| Trajectron++ | 1.52 | 2.80 | 4.53 | 1.06 | 2.06 | 3.46 |

In Tab.9, we show the importance of the map information to the discriminator by removing the map input to the discriminator and keep all the rest parts the same (the map input to the generator is kept). Due to the lack of map information, the discriminator cannot be well trained and the performance drops compared to our full model. Models are evaluated on nuScenes dataset.

# D    QUALITATIVE RESULTS

Here, we demonstrate qualitative results of our method compared with Trajectron++ for 4 seconds prediction, trained for 3 seconds prediction. We randomly sample 50 trajectories from model for each prediction, use kernel density estimation to approximate the total output distribution from the samples, and print it out in Fig.3. The ground truth trajectories are represented by white points. Compared to Trajectron++, our method reduces the uncertainty of the future by a large margin and also increase the accuracy.

The classified latent intent behavior helps us to widen the difference among trajectories from different behavior intents. To demonstrate this, we plot out trajectories for every latent intents, totally 25 intents including the unlikely latent intents given the input data, for both our method and Trajectron++ in Fig.6. The white points are ground truth trajectories. The red trajectories are the behaviors $\hat{\mathbf{z}}$ with at least 5% probability ($p(\hat{\mathbf{z}}|\mathbf{x}, \mathbf{m}) \geq 0.05$). The gray trajectories are behaviors which are less possible to occur ($p(\hat{\mathbf{z}}|\mathbf{x}, \mathbf{m}) < 0.05$). From the visualization we can see that the latent behaviors in our method are more diverse and distinguishable compared to Trajectron++.

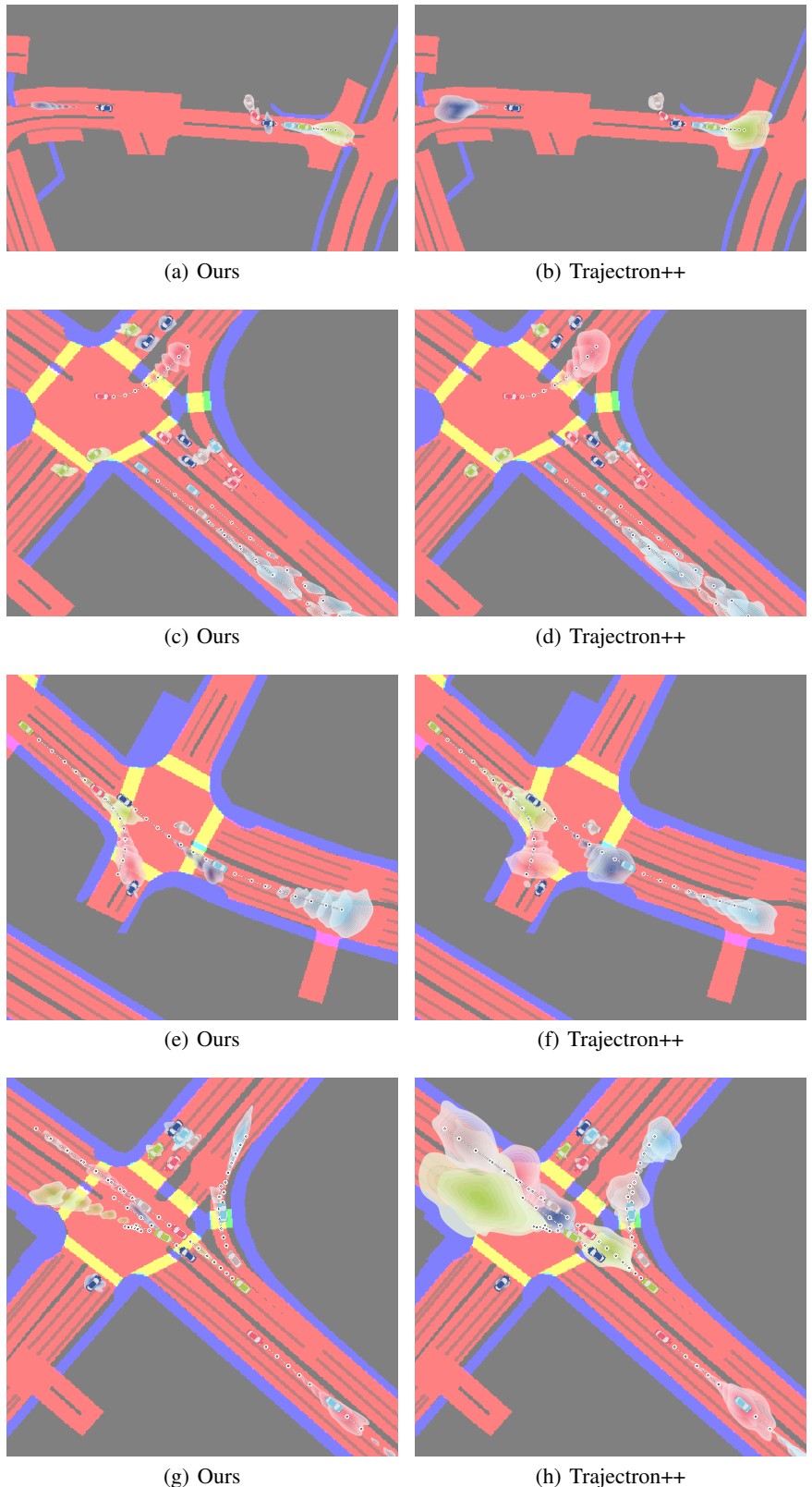

(a) Ours        (b) Trajectron++

(c) Ours        (d) Trajectron++

(e) Ours        (f) Trajectron++

(g) Ours        (h) Trajectron++

Figure 3: Qualitative results of our method and Trajectron++. Compared to Trajectron++, our method significantly reduces the uncertainty of the prediction in all scenes with improved accuracy. White points denotes the ground truth trajectories.

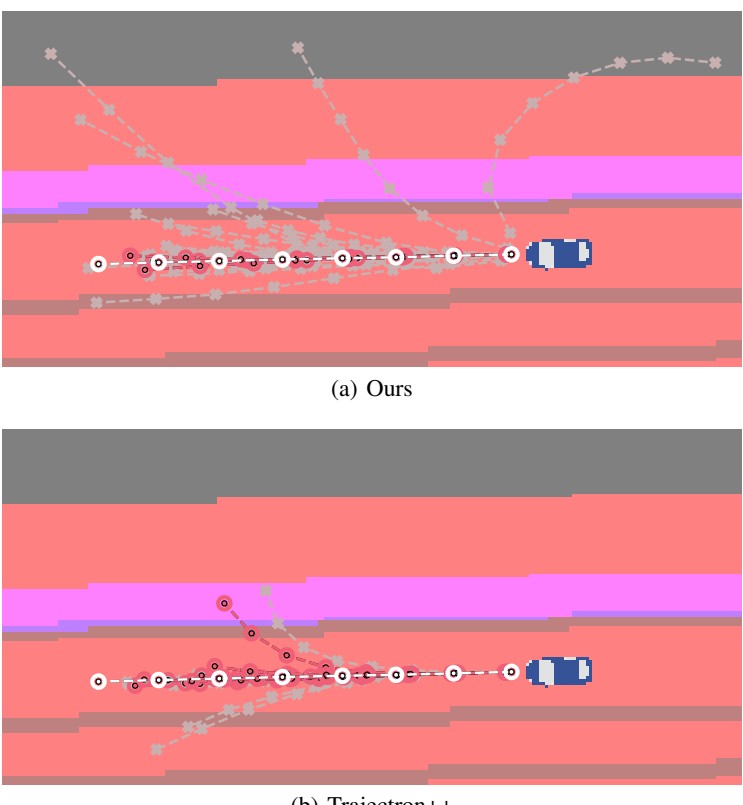

(a) Ours

(b) Trajectron++

Figure 4: The trajectories from all behavior intents generated by our method and Trajectron++. We force the model to predict trajectory for all behaviors no matter the behaviors are possible judged by the model or not. White points denote the ground truth trajectories. The other points denote predicted trajectories with different behavior intents. With the help of classified latent intent behavior, we obtain more diverse behaviors compared to Trajectron++. Note that red points comes from the intents which are likely under the judgement of models given the input data. The gray points comes from intents which are very unlikely to happen and we forcibly set it for demonstration. Note that Trajectory++ predicts unsafe trajectory with a high likelihood. While Our method have a capability to predict diverse trajectories but unsafe modes have a vert low likelihood

# E    PEDESTRIAN DATASETS

Here, we train our model on the pedestrians dataset ETH (Pellegrini et al., 2009) and UCY (Leal-Taixé et al., 2014) without map information. A leave-one-out technique is used for evaluation, similar to previous work (Alahi et al., 2016; Gupta et al., 2018; Ivanovic & Pavone, 2019; Kosaraju et al., 2019; Sadeghian et al., 2019; Salzmann et al., 2020), where the model is trained in four datasets and tested in the fifth dataset. The assessment uses an observation period of 8 timesteps (3.2s) and a projected horizon of 12 timesteps (4.8s). Note that different from experiments in nuScenes dataset, our model is trained from scratch here.

We show in table 10 our performance on the UCY datasets. In addition, the model's deterministic ML output scheme is used, which produces the most likely single trajectory of the model. With only using the the notion of classified intents and hallucinated intents that can be captured by a discrete latent vector $\hat{\mathbf{z}}$, we see a slight improvement in the FDE results with almost 7% over Trajectron++.

Table 10: **Without Map.** The performance of our method on UCY pedestrian datasets. We don't have map information in this experiments. Although, our method still achieve comparable performance compared to other state-of-the-art methods and get the best performance in the FDE of most likely trajectories averaged over 3 datasets. Lower is better. Bold indicates best. Our method is significantly better.

| | ADE/FDE ML | | | |
|---|---|---|---|---|
| **Dataset** | S-LSTM (Gupta et al., 2018) | S-ATTN (Vemula et al., 2018) | Trajectron++ (Salzmann et al., 2020) | Ours |
| Univ | 0.67/1.40 | 0.33/3.92 | 0.41/1.07 | 0.42/1.07 |
| Zara 1 | 0.47/1.00 | 0.20/0.52 | 0.30/0.77 | 0.27/0.67 |
| Zara 2 | 0.56/1.17 | 0.30/2.13 | 0.23/0.59 | 0.20/0.52 |
| Average | 0.57/1.19 | **0.28**/2.19 | 0.31/0.81 | 0.29/**0.75** |

# F   DETAILED ALGORITHM

---

**Algorithm 2:** Detailed Training Process

---

Predicted horizon $T$, Integration model $Inte$

Initialize $Enc_{\theta_E}, Dec_{\theta_D}, Dis_\phi$;

Initialize learning rate $\alpha, \beta$;

**while** *not converge* **do**

   // discriminator

   $\mathbf{y}_{gt}, \mathbf{x}, \mathbf{m} \sim$ Dataset

   // Compute the high level features $\mathbf{e}$ and the distribution of latent code $\mathbf{z}$.

   $\mathrm{P}(\mathbf{z}|\mathbf{x}, \mathbf{m}), \mathbf{e} = Enc_\theta(\mathbf{x}, \mathbf{m})$

   // Sample the classified intent $\hat{\mathbf{z}}$

   $\hat{\mathbf{z}} \sim \mathrm{P}(\mathbf{z}|\mathbf{x}, \mathbf{m})$

   **for** *t in range(T)* **do**

      $\mathrm{P}(a_t|\hat{a}_{t-1}, \mathbf{x}, \mathbf{m}) = Dec_{\theta_D}(\mathbf{e}, \hat{\mathbf{z}}, \hat{a}_{t-1})$

      $\hat{a}_t \sim \mathrm{P}(a_t|\hat{a}_{t-1}, \mathbf{x}, \mathbf{m})$

   **end**

   // convert the action into trajectories by integration model

   $\hat{\mathbf{y}} = [\hat{y}_1, \hat{y}_2, ...\hat{y}_T] = Inte(y_0, \hat{a}_1, \hat{a}_2, ..., \hat{a}_T)$

   // Discriminator judge whether the given trajectory is real/fake and classified to which $\mathbf{z}$

   // $\tilde{\mathbf{z}}$ represents the classification result $\mathrm{P}(\mathbf{z}|\hat{\mathbf{y}}, \mathbf{m})$. The $i$-th element $\tilde{z}_i$ is the probability $\hat{\mathbf{y}}$
    belongs to $i$-th $\mathbf{z}$ judged by the discriminator

   $D(\hat{\mathbf{y}}), \tilde{\mathbf{z}} = Dis_\psi(\hat{\mathbf{y}}, \mathbf{m})$

   $D(\mathbf{y}_{gt}), \_ = Dis_\psi(\mathbf{y}_{gt}, \mathbf{m})$

   $L_C = CrossEntropy(\hat{\mathbf{z}}, \tilde{\mathbf{z}}) = -\sum_i \hat{z}_i \log \tilde{z}_i$ // classification loss

   $L_D = (1 - D(\mathbf{y}_{gt}))^2 + (D(\hat{\mathbf{y}}))^2 + L_c$ // GAN loss plus classification loss

   $\phi = \phi - \alpha\nabla_\phi L_D$

   // generator

   $\_, \mathbf{x}, \mathbf{m} \sim$ Dataset

   $\mathrm{P}(\mathbf{z}|\mathbf{x}, \mathbf{m}), \mathbf{e} = Enc_\theta(\mathbf{x}, \mathbf{m})$

   // Sample $\hat{\mathbf{z}}, \hat{\mathbf{z}}'$ and a time step $t_h$. Their combination is viewed as the hallucinated intent $\hat{\mathbf{z}}_h$

   $\hat{\mathbf{z}}, \hat{\mathbf{z}}' \sim \mathrm{P}(\mathbf{z}|\mathbf{x}, \mathbf{m}), \hat{\mathbf{z}}' \neq \hat{\mathbf{z}}$

   $t_h \sim Uniform(2, T - 1)$

   // Generate hallucinated trajectory $\hat{\mathbf{y}}_h$

   **for** *t in range($t_h$)* **do**

      $\mathrm{P}(a_{h,t}|\hat{a}_{h,t-1}, \mathbf{x}, \mathbf{m}) = Dec_{\theta_D}(\mathbf{e}, \hat{\mathbf{z}}, \hat{a}_{h,t-1})$ // Conditioned on $\hat{\mathbf{z}}$

      $\hat{a}_{h,t} \sim \mathrm{P}(a_{h,t}|\hat{a}_{h,t-1}, \mathbf{x}, \mathbf{m})$

   **end**

   **for** *t in range($t_h + 1, T$)* **do**

      $\mathrm{P}(a_{h,t}|\hat{a}_{h,t-1}, \mathbf{x}, \mathbf{m}) = Dec_{\theta_D}(\mathbf{e}, \hat{\mathbf{z}}', \hat{a}_{h,t-1})$ // Conditioned on $\hat{\mathbf{z}}'$

      $\hat{a}_{h,t} \sim \mathrm{P}(a_{h,t}|\hat{a}_{h,t-1}, \mathbf{x}, \mathbf{m})$

   **end**

   $\hat{\mathbf{y}}_h = [\hat{y}_{h,1}, \hat{y}_{h,2}, ...\hat{y}_{h,T}] = Inte(y_0, \hat{a}_{h,1}, \hat{a}_{h,2}, ..., \hat{a}_{h,T})$

   $D(\hat{\mathbf{y}}_h), \tilde{\mathbf{z}}_h = Dis_\psi(\hat{\mathbf{y}}_h, \mathbf{m})$

   // Make $\hat{\mathbf{y}}_h$ hard to be classified by reducing the cross entropy between a uniform
    distribution and the classification results $\tilde{\mathbf{z}}_h$. $N$ is the number of latent code

   $L_h = CrossEntropy(UniformDist, \tilde{\mathbf{z}}_h) = -\sum_i \frac{1}{N} \log \tilde{z}_{h,i}$

   $L_{G,c} = (D(\hat{\mathbf{y}}) - 1)^2 + L_c$

   $L_{G,h} = (D(\hat{\mathbf{y}}_h) - 1)^2 + L_h$

   $\theta = \theta - \beta\nabla_\theta(\lambda L_{G,c} + (1 - \lambda)L_{G,h})$

   $\mathbf{y}_{gt}, \mathbf{x}, \mathbf{m} \sim$ Dataset

   $\theta = \theta - \beta\nabla_\theta(L_{\text{traj++}})$ // Original Trajectron++ loss

**end**

---

# G   CODE PATCH FOR TRAINING LOOP

```
1  ##################################
2  #             TRAINING            #
3  ##################################
4
5  pbar = tqdm(data_loader, ncols=80)
6  pbar_gen = iter(pbar)
7
8  d_loss = g_loss = train_loss = torch.zeros(1)
9  not_empty = True
10 while not_empty:
11     trajectron.set_curr_iter(curr_iter)
12     trajectron.step_annealers(node_type)
13
14     # -------------- discriminator --------------#
15     # 1. Compute the high level featurese
16     # and the distribution of latent codez.
17     # 2. Sample the classified intent z
18     # 3. convert the action into trajectories
19     # by integration model
20     # 4. Discriminator judge whether the given
21     # trajectory is real/fake and classified to which z
22     # 5.   z  represents the classification result P(z| y ,m).
23     # The i-th element   zi  is the probability  y
24     # belongs to i-th z judged by the discriminator
25     optimizer_d.zero_grad()
26     part2train(model_registrar, "discriminator")
27
28     batch, not_empty = fetch_batch(pbar_gen)
29     if not not_empty:
30         break
31
32     d_loss, dc_loss, d_real, d_fake = trajectron.gan_d_loss(
33         batch, node_type, grid_std=args.grid_std, grid_max=args.grid_max
34     )
35     loss = d_loss + args.class_lambda * dc_loss
36     (args.loss_weight_total * loss).backward()
37     optimizer_d.step()
38
39     # -------------- Generator --------------#
40     # 1. Sample additional latent code    z    and
41     # a time step th to assemble hallucinated intent   zh
42     # 2. Generate hallucinated trajectory  yh
43     # 3. Make  yh  hard to be classified by
44     # reducing the cross entropy between
45     # a uniform distribution and the classification results   zh .
46     optimizer.zero_grad()
47     part2train(model_registrar, "generator")
48
49     batch, not_empty = fetch_batch(pbar_gen)
50     if not not_empty:
51         break
52
53     g_loss, c_loss = trajectron.gan_g_loss(
54         batch, node_type, grid_std=args.grid_std, grid_max=args.grid_max
55     )
56     (
57         args.real_ratio * args.g_factor * (g_loss + args.class_lambda *
58     c_loss)
58     ).backward()
59     g_loss, creative_loss = trajectron.gan_g_loss(
60         batch,
61         node_type,
62         grid_std=args.grid_std,
```

```
63          grid_max=args.grid_max,
64          creative=creative_mode,
65      )
66      loss = (
67          (1 - args.real_ratio)
68          * args.g_factor
69          * (g_loss + args.creative_lambda * creative_loss)
70      )
71      (args.loss_weight_total * loss).backward()
72
73      optimizer.step()
74
75      # ---------- trajectron++ ----------
76      optimizer.zero_grad()
77      part2train(model_registrar, "generator")
78
79      batch, not_empty = fetch_batch(pbar_gen)
80      if not not_empty:
81          break
82      train_loss = trajectron.train_loss(batch, node_type)
83      train_loss.backward()
84
85      optimizer.step()
86
87      # Stepping forward the learning rate scheduler and annealers.
88      lr_scheduler.step()
89      lr_scheduler_d.step()
90
91      curr_iter += 1
```

Listing 1: Training Loop

