# OpenReview forum: "HalentNet: Multimodal Trajectory Forecasting with Hallucinative Intents"
_ICLR.cc/2021/Conference — ICLR 2021 Poster_

### Official Review · AnonReviewer3 · 2020-10-20
**Interesting hallucination technique for data augmentation**

**Rating:** 8
**Confidence:** 3

**Review:**

This paper presents an interesting technique to generate multimodal trajectory GAN and a carefully designed latent intent space. This latent intent space allows an operation termed as hallucination, which switches agent intents to enrich the latent spaces' coverage.

The paper is overall clear and well-written. However, there is one point regarding the hallucinative learning probably can benefit from more elaboration: the time step "t" is only introduced in the Hallucinative Learning paragraph. How is the time step being used in the network? Is the network only predicting y_{gt} as a simple waypoint (t=1), or a sequence of waypoints (a trajectory, in that case, what is the horizon)?  If I understand correctly, the main hallucination idea is to extract as many intentions in the latent space as possible, from the training data, and then the hallucination is implemented as randomly switching between different learned intents to generate augmented data? This point needs to be carefully clarified.

Regarding the idea of using "hallucination" to generate more training data, many recent works have used the idea of hallucination for data augmentation, or even generating training data from scratch. For example, https://arxiv.org/pdf/2010.08098.pdf and https://arxiv.org/pdf/2007.14479.pdf generate training data using hallucination based on geometric feasibility, instead of latent intent, https://openaccess.thecvf.com/content_CVPR_2019/papers/Zhang_Few-Shot_Learning_via_Saliency-Guided_Hallucination_of_Samples_CVPR_2019_paper.pdf uses saliency to guide the generation of hallucination, https://openaccess.thecvf.com/content_CVPR_2020/papers/Li_Adversarial_Feature_Hallucination_Networks_for_Few-Shot_Learning_CVPR_2020_paper.pdf also used a very similar adversarial approach to generate feature hallucination. These works can help to set stage in the related work section.

---

> ### Author Response · Authors · 2020-11-20
> **Answers to the Questions**
>
> We thank Reviewer 3 for the valuable feedback reviews. We address here the questions, and we incorporated all the feedback.
>
> ***Q1 How is the time step being used in the network? Is the network only predicting $y_{gt}$ as a simple waypoint (t=1), or a sequence of waypoints (a trajectory, in that case, what is the horizon)? ***
> $y_{gt}$ is a sequence of waypoints. For the classified intent, we keep the latent variable z the same over the whole predicted horizon. For the hallucination, we switch latent z once at a randomly sampled timestep $t_h$ during training. We have updated the hallucinative learning subsection in the method section to highlight the time step for better clarity. In addition, we revised our detailed algorithm in Appendix G to better show how the time step is used in our method.
>
> ***Q2 Additional Related Work***
> Thanks for the pointers! We included the suggested references in the revised version to enrich our related work section.

---

### Official Review · AnonReviewer1 · 2020-10-26
**R1 updated review after author feedback**

**Rating:** 5
**Confidence:** 5

**Review:**

Summary: The authors tackle the problem of trajectory prediction in autonomous vehicles.  They propose a data augmentation scheme where they enrich the raw trajectories with synthetically generated trajectories to reduce spurious modes in the predictions. This enriched dataset is hoped to improve  accuracy in trajectory prediction problems.

Strengths:

1. The paper belongs to an active area of research. Reliable and accurate Trajectory prediction is one of the central problems in autonomous driving.

Weaknesses:

1. The paper is very hard to read. For example, the contributions are confusing and not clearly defined. In the first paragraph of the Introduction, a data augmentation scheme is presented. But then on Pg.2, the authors present two notions of intents as their contributions. What is the relation? From my understanding, the authors use $\hat z \sim P(z)$ to generate new data  (I'm looking at  Eqn 4). And this eqn 4 is termed as the "classified intent". There are several things to unpack here:
First, these contributions are not well-defined. What is a "classified intent" and "hallucinative intent". These are not standard terms in the trajectory prediction literature. So these terms need to be defined and explained. I would like to see a mathematical definition and references to relevant citations in the traffic psychology literature.
Once it is formally defined what a "classified intent" and "hallucinative intent" is, the next thing would be to formally derive equations 4 and 5 from those definitions. Because eqns 4/5 are not derived with supporting derivations and justifications, the main concern here is that I don't find the motivation or relevance of these equations convincing.
My suggestion here is that most of the current material on Pg.4 can be moved to an implementation section. Instead, use that space to include the motivations, definitions, justifications, and derivations for eqns 4/5 as explained above.

2. The questionable nature of the proposed equations (4, 5) directly relates to my next point, that is, the lack of any useful improvement over SOTA. In Tab. 1, the max FDE improvement is 18cm. In Tab 2. it is 5cm. Furthermore, the authors report gains in terms of percentages, which is misleading and even dangerous, since in the real world, it is important to know absolute errors and not percentage errors relative to another benchmark.

3. Additionally, the authors primarily compare using the FDE (a weaker and insufficient metric than the ADE) on most occasions, most critical of which are Tab.1 which contains the comparison with other methods and Tab.5 which contains the ablation experiments that highlight the benefits of the proposed equations 4 and 5. For fair evaluation, it is necessary to also present comparisons using ADE.

In summary, the paper suffers from lack of a clear justification of the proposed contributions, unfair evaluations, and questionable significance of the results

---

> ### Author Response · Authors · 2020-11-20
> **Part 1: Classified and Hallucinated Intents**
>
> We thank Reviewer 1 for the feedback. We find the review does not give credit to several efforts in the paper based on which we think there could be a misunderstanding.
>
> ***Q1 The paper is very hard to read***
>
> According to Reviewer 2, 3, and 4, our paper is well-written and easy to understand. Trying to do what is within our hands to further improve clarity, (1) we revised our introduction section, (2) clarified more the difference between classified and hallucinated intents in the Classified latent Intent Behavior and Hallucinative Learning subsections, and (3) attached a detailed algorithm in Appendix G.
>
> ***Q2 In the first paragraph of the Introduction, a data augmentation scheme is presented. But then on Pg.2, the authors present two notions of intents as their contributions. What is the relation? ***
>
> The relation is that our hallucinated data augmentation scheme is done by combinations of classified intents to our model. So, our augmentation operates in latent discrete spaces that represent intents/behavior.
> The new combination of intentions here means the proposed hallucinated intent. The other one, the classified intent, is needed for the hallucinated intent to work properly, discussed in the ablation study subsection. We revised our introduction section to improve clarity and transition between paragraphs (updates are in blue).
>
>
> ***Q3 What is a "classified intent" and "hallucinated intent". These are not standard terms in the trajectory prediction literature.***
>
> This is true since these terms define our contribution and novelty. The term driving intent, however, is commonly used in the literature and several recent papers modeling the driving behavior as a discrete latent variable (Salzmann et al., 2020; Tang \& Salakhutdinov, 2019). We answer the definition here, but we refer Reviewer 1 to Classified Latent Intent Behavior subsection, Hallucinative Learning subsection, and the detailed algorithm in Appendix G in the paper (added in this version )  for more details. Below is copied from the current version of the paper.
> **Classified Intent.**
> When motion is predicted, typically one fixed one-hot latent code $z$ is set for the entire trajectory. Our model (both generator and discriminator) is trained by the classification loss $L_c$ (Equation 4) to encourage that the trajectory generated in this case can be classified with a high confidence score to the latent code $z$ it is from. We call this latent code classified intent. $L_c$ encourages these intents to be more distinguishable from each other.
> **Hallucinated Intent.**
> The goal of hallucinated intent is to augment the behaviors by combining 2 different latent codes in the temporal dimension to generate augmented trajectories. More concretely, the trajectories generated by hallucinated intent are first conditioned on the first one-hot latent code to time step $t_h$. They are switched to the second latent code in the remaining steps.
> These generated trajectories from this hallucinated intent variable are encouraged to be hard to be classified into any of the one-hot latent code by our hallucinative loss $L_h$ (Equation 5).
> This means the trajectories from the hallucinated intents are different from those from the classified intent as judged by our discriminator.
>
> We updated a detailed algorithm to show how the classified intent and hallucinative intent are generated and used in Appendix G.
>
> ***Q4 What justifies our $L_h$ design choice. ***
> We design our hallucinative learning loss $L_h$ as the cross entropy between a uniform distribution and the classification results. In addition to this design choice (let's call as the default $L_h$), we also experiment with the following 2 additional choices and attached the results in Appendix C Table 8 of the revised paper.
> **Hallecinative Class Intent $L_h (N+1)$.** Where the model classifies the hallucinated trajectory as a new class $N+1$ that is orthogonal to all classified intents.
> **Hallecinative Mixed Intent $L_h (mixup)$.** Where the model classifies the hallucinated trajectory as a Mixup between the two sampled intents.
>
> Ablation results in Table 8 show that our original design choice, $L_h$ (default),  has the best performance. $L_h (mixup)$ also achieves comparable performance (Performance drops about 5\% at the 5-second FDE-Full case). However, $L_h (N+1)$ performs poorly (Performance drops about 30\% at the 5-second FDE-Full case). We think the reason is that $L_h (N+1)$ offers a contradict learning signal. $L_h (mixup)$ performs better but seems harder to train compared to the default $L_h$.

---

> ### Author Response · Authors · 2020-11-20
> **Part 2  Our Results'  Significance and Other Questions**
>
> ***Q5 The lack of any useful improvement over SOTA. Furthermore, the authors report gains in terms of percentages, which is misleading.***
>
>  We respectfully disagree. The FDE improvement is 18cm in Table 1 (vehicles) and 5cm (pedestrians) in Table 2, the improvement is still about 8\%. The significance of our method becomes more apparent on a longer prediction horizon as we show in Table 3.  In the 6-second prediction case, we reduce FDE-Full from 6.70 to 4.93. This is 1.76 meters, and the improvement is about 27\%.
> Because the absolute value is reported, we don't agree that we are misleading something. Besides, the generated trajectories from our method have a clear higher quality compared to our baseline measured by human study shown in Appendix D. We obtain 39.6\% more votes than our baseline. In addition, our method reduces the road boundary violation rate by a large margin from 19\% to 9\% reported in Table 3. This is also a significant improvement.
>
>
> ***Q6 Equation 4/5 are not derived with supporting derivations and justifications. ***
>
> Equation 4 is the classification cross entropy loss between the one-hot ground truth (The classified intent $\mathbf{\hat z}$ we use to generate the trajectory $\mathbf{\hat y}$) and the predicted categorical distribution (we use vector $\mathbf{\tilde z}$ to represent this distribution, where the $i$-th element of $\mathbf{\tilde z}$, $\tilde z_i$, is the probability that $\mathbf{\hat y}$ belongs to $i$-th latent).
> Therefore, the cross entropy between $\mathbf{\hat z}$ and $\mathbf{\tilde z}$ is Equation 4 $L_c = - \sum_i \hat z_{i} \log \tilde z_i$
>  We train our model using Equation 4 to classify the given trajectory y into one of the latent code z. The motivation, as we mentioned in the subsection classified latent intent behavior, is that the decoder will widen the difference of y conditioned on different z to reduce the classification difficulty, and therefore make different z more distinct from each other.
>
> Regarding Equation 5, as we mentioned in the subsection Hallucinative learning, we trained the generator with Equation 5 when hallucinated intent is applied so that the trajectory $\mathbf{\hat y_h}$ generated by hallucinated intent is difficult to be classified into any latent code. This encourages the hallucinated trajectory to be different from the trajectory from classified intent.
> Similar to Equation 4, Equation 5 is also a cross entropy loss, but is between a discrete uniform distribution and the predicted result  $\mathbf{\tilde z}$. Assume we have $N$ different latent code, the probability for each latent code in a uniform distribution is $\frac{1}{N}$. Therefore, the cross entropy is $L_{h} = - \sum_i \frac{1}{N} \log \tilde z_i$, which is our Equation 5.
>
> ***Q7 For fair evaluation, it is necessary to also present comparisons using ADE.***
>
> Thank you for your suggestion. We have attached a new table 6 in Appendix C to report the ADE scores. We reduce ADE-Full in the 4-second prediction case from 1.16 to 0.91 compared to Trajectron++ (Last row). The relative improvement is 21.5\%.  The results of ADE are consistent with our statement in the main paper.

---

> ### Author Response · Authors · 2020-11-21
> **Thanks for the score increase**
>
> We appreciate a lot the score increase by Reviewer1. If there is something R1 think that we can do to further improve, please let us know and we will do what is within our hands. Thanks a lot!

---

### Official Review · AnonReviewer4 · 2020-10-29
**Solid paper for consideration of accept**

**Rating:** 6
**Confidence:** 3

**Review:**

Summary:
The paper designed a framework for motion forecasting (trajectory prediction), with emphasis on multimodal distribution modeling and generalization. Specifically, they use latent code to model agent's intents. This latent code combined with historical trajectories and map were used to generate future trajectories, which were further judged by a discriminator. Besides, they added latent code classification and hallucinative data augmentation for performance boosting.

Reasons for score:
The major reason for the voting of accept is that this paper is solid and generally easy to understand. The interplay of latent code classification and hallucinative is interesting.

Pros:
1. Solid method development, fluent method introduction, and systematic ablation studies.
2. The proposed method seems to be interesting, especially the adding of latent code z to generate trajectories and then classify them.

Cons:
1. The proposed methods seems to be combination of existing components, which limits its theoretical contributions.
2. The underlying reasons for the success of different components (classification of latent intent and hallucinative latent intent) are hard to explain.

Please address the following questions:
1. It would be interesting to see how this method compare with "Mercat, Jean, et al. "Multi-head attention for multi-modal joint vehicle motion forecasting." 2020 IEEE International Conference on Robotics and Automation (ICRA). IEEE, 2020." which seems to be a state of art.
2. During inference, how do you generate and select multiple trajectory candidates?
3. It seems like only when both classification latent and hallucinative latent are combined can they improve the performance. I notice the authors discussed about this, but I think this interplay needs more discussion.
4. Did you try to see what is the contribution of map?
5. For equation (2), it is better to display it as multi lines for easy understanding. Also, P(z,|x,m),e = Encθ(x,m), the "," after z seems to be a typo.

---

> ### Author Response · Authors · 2020-11-20
> **Answers to the Questions**
>
> We thank Reviewer 4 for the valuable and helpful comments. We incorporated all the feedback here.
>
> ***Q1 It would be interesting to see how this method compares with "Mercat, Jean, et al. "Multi-head attention for multi-modal joint vehicle motion forecasting."***
>
> Thank you for your suggestion! We were not able to find the code publicly available. We reached out to the authors asking if there is a possibility to share the code, but we did not hear back. We cited the paper to enrich the related work section.
>
> ***Q2 During inference, how do you generate and select multiple trajectory candidates?***
>
> In the FDE/ADE-ML case, a greedy search is used step by step to generate the trajectory. That means we first select the latent z with the highest probability, then the predicted mean at each time step is picked up as the output. In other cases, the trajectory candidates are randomly sampled.
>
> ***Q3 It seems like only when both classification latent and hallucinative latent are combined can they improve the performance. I think this interplay needs more discussion. ***
>
> As we mentioned in the introduction (updated), our method benefits from the hallucinated agent behaviors as an implicit data augmentation approach.
> **Role of $L_c$** The loss $L_c$ for classified intent is trained to make the trajectories generated from different $z$ more diverse. And the augmented behaviors are generated by hallucinative learning. This diversity helps the hallucinated loss which operates as an exploratory learning signal on the space of transitions between classified intents over time.
> **Removing $L_h$** If we remove the loss $L_h$ for hallucinative learning and only keep $L_c$, our implicit data augmentation doesn't exist anymore.
> In contrast, hallucinative learning loss $L_h$ is trained to generate trajectories that cannot be classified into any latent code by the discriminator when we use hallucinated intent.  If we remove $L_c$ and only keep $L_h$, our discriminator doesn't learn to classify trajectories making the loss $L_h$ meaningless. Therefore, the interplay between these 2 methods is necessary for our method. We updated our ablation subsection to enrich the discussion.
> ***What justifies our $L_h$ design choice. ***
> We design our hallucinative learning loss $L_h$ as the cross entropy between a uniform distribution and the classification results. In addition to this design choice (let's call as the default $L_h$), we also experiment with the following 2 additional choices and attached the results in Appendix C Table 8 of the revised paper.
> **Hallecinative Class Intent $L_h (N+1)$.** Where the model classifies the hallucinated trajectory as a new class $N+1$ that is orthogonal to all classified intents.
> **Hallecinative Mixed Intent $L_h (mixup)$.** Where the model classifies the hallucinated trajectory as Mixup between the two sampled intents.
> Ablation results in Table 8 show that our original design choice, $L_h$ (default),  has the best performance. $L_h (mixup)$ also achieves comparable performance (Performance drops about 5% at the 5-second FDE-Full case). However, $L_h (N+1)$ performs poorly (Performance drops about 30% at the 5-second FDE-Full case). We think the reason is that $L_h (N+1)$ offers a contradict learning signal. $L_h (mixup)$ performs better but seems harder to train compared to the default $L_h$.
>
>
>
> ***Q4 Did you try to see what is the contribution of map?.***
>
> Thank you for your suggestion! Our method belongs to the class of methods that uses the map information similar to (Salzmann et al., 2020; Tang & Salakhutdinov, 2019; Wang et al., 2020). However, we also ran new experiments to verify the contribution of the map. We remove the map input to the discriminator on the nuScenes case. Results comparing this no-map version, our full version, and the original Trajectron++ are shown in Table 7 of the updated paper.  Note that only the map input to the discriminator is removed in the no-map version. The generator of our full version, no-map version, and the original Trajectron++ still receive maps as part of the input. The performance drops by about 10\% when we remove the map input to the discriminator, which indicates that the map information is important for training a good discriminator and therefore is important to our performance. This is consistent with our statement in the UCY dataset.
>
> ***Q5 For equation (2), it is better to display it as multi-lines for easy understanding. Also, the "," after z seems to be a typo.***
>
> Thank you for your suggestion! We have updated the original equation 2 in the new version.

---

> ### Author Response · Authors · 2020-11-21
> **Part 2 Method Novelty and Results' Significance**
>
> ***Q6 The proposed methods seem to be a combination of existing components, which limits its theoretical contributions.***
>
> To our knowledge, our proposed classified intent and hallucinated intents are conceptually new, and hence we believe our method to model them is novel. From the results perspective, the significance of our hallucinated augmentation signal becomes more apparent on a longer prediction horizon, as we show in Table 3. In the 6-second prediction case, we reduce FDE-Full from 6.70 to 4.93. This is 1.76 meters, and the improvement is about 27%. Besides, the generated trajectories from our method have a clear higher quality compared to our baseline measured by human study shown in Appendix D. We obtain 39.6% more votes than our baseline

---

### Official Review · AnonReviewer2 · 2020-11-02
**Review [Updated]**

**Rating:** 6
**Confidence:** 4

**Review:**

**SUMMARY**

The present work considers the problem of multi-agent trajectory prediction. Its main contribution is incorporating generative augmentation losses for improving the quality of a trajectory predictor. This is achieved by allowing trajetcory predictors to model intent as an unobserved latent variable in the model and using this to generate trajectories corresponding to different intentions. The work also proposes to use a descriminative loss encouraging diversity of the intents and an additional "hallucination" loss that allows for modelling mixed intents.

The idea is integrated with Trajectron++ and it is shown how the newly approach improves upon the Final Displacement Error, a comon trajectory prediction metric. In human experiments, it is also shown that the trajectories predicted by the new approach are considered more realistic by humans in comparison to the baseline.

**STRENGTHS**
- The idea of a structured generative loss for creating a richer diversity of predictions has a lot of potential to improve the trajectory prediction tasks.
- Human studies seem to be a very interesting idea for studying trajectory prediction.

**WEAKNESSES**
- Although the authors claim somewhat broad applicability of their approach, the idea is ultimately only demonstrated on Trajectron++
- The evaluations require some more data to be better interpretable.

**CLARITY**

I found this work mostly clear to read. It sometimes makes an assumption about the reader's familiarity with more specialized techniques such as LSGAN and spectral normalization. While there is probably no space to introduce everything used by the authors and I do not hold it against them, maybe it is worth briefly motivating these and other non-obvious design choices without going much into detail. Further smaller clarity questions/remarks are:
- What is the motivation behind the discreteness of the latent variable?
- What do the authors mean by "unsupervised discrete random variables"? As I understand it, a random variable is a mathematical object that typically does not entail properties such as supervised or unsupervised.  Maybe simply write "discrete latent variable"?

**REPRODUCIBILITY**

I believe the work to be mostly reproducible. It is not possible to judge the usefulness of the code without it being available during the review process. I still applaud the author's intent to release it.

**EVALUATION**
- Focusing the evaluation on one predictor (Trajectron++) seems insufficient given that the authors make a much broader claim about the general usefulness of their approach.
- Similarly, the claim that the proposed approach outperforms multiple state-of-the-art predictors seems misleading. In Table 1, Trajectron would have been already the clearly best model even without the addition proposed by the authors.  In Table 6, the proposed approach does not always outperform the baseline and does not even always outperform Trajectron++. While I do not believe that every paper has to outperform all previous papers always on all metrics, it would be interesting to see a discussion/analysis on why this happens.
- Also why does Table 6 only contain UCY? Is it because ETH is already shown in Table 4? The comparisons seem different to me, so maybe worth also integrating ETH in table 6?
- Because the proposed method's performance is still somewhat close to Trajectron++'s, it would be interesting to see several runs of the same experiments also reporting the standard deviation of the obtained metrics. At least for the comparison with Trajectron++.

**SUMMARY**

In summary, I think the authors propose an interesting idea and I believe using humans to judge the quality of trajectories is a cool evaluation methodology. The paper's weak point is a much broader claim while it is only evaluated with Trajectron++. And, in the comparison with this baseline, it is not clear to what extent the work will constantly outperform it. That being said, I want to encourage the authors in following this line of work and providing further evaluations.

**POST-DISCUSSION UPDATE**

I believe the authors to have addressed some of my concerns and I appreciate the additional experiments demonstrating that even the close results were more than a mere statistical artifact. Some other points remain still open such as the limited focus on Trajectron in evaluations.  In summary, I believe that the paper has now surpassed the acceptance threshold and am happy to recommend its publication.

---

> ### Author Response · Authors · 2020-11-20
> **Answers to Questions**
>
> Thank you for your detailed review, questions and suggestions. We here address them and incorporated all the feedback.
>
> ***Q1 What is the motivation behind the discreteness of the latent variable?***
>
> Discrete latent variables are used in previous works like MFP [1] and Trajectron++ [2]. In line with these works, we think human driving behavior/ intent is closer to discrete than continuous. (e.g. turn left, go straight, speed up). Hence, using discrete latent is intrinsically suitable to model human behavior intent. It also increases the interpretability since we can visualize the trajectories from different z to show the behavior we learn. Lastly and more importantly, modeling hallucinative deviation signals on discrete random variables is simpler and easier to train and hence is more practical. We performed different ablations to justify our particular design of $L_h$ and why other choices are less effective in Table 8 of Appendix C.
>
> ***Q2 What do the authors mean by "unsupervised discrete random variables"? As I understand it, a random variable is a mathematical object that typically does not entail properties such as supervised or unsupervised. Maybe simply write "discrete latent variable"?***
>
> ''unsupervised'' here means that these discrete random variables are latent and do not require behavior/intent labels to be trained. We have revised our paper to incorporate your comments.
>
> ***Q3 Why does Table 6 only contain UCY?***
>
> In table 6, models are trained without map information. Since our method relies highly on the quality of the discriminator,  the map information is important for a well-trained discriminator, and ETH offers simple maps for us to use, we evaluate ETH with map information in table 4. We don't use the leave-one-out technique for evaluation here due to the lack of UCY maps. Therefore, Table 4 is evaluated in a different style. The method we use on UCY experiments (Table 6) is more like a variant of our approach due to the lack of map information. To verify the map information's importance to the discriminator, we add a new ablation study in Table 7. Details are further discussed in the next question.
>
>
> ***Q4 The claim that the proposed approach outperforms multiple state-of-the-art predictors seems misleading. It is not clear to what extent the work will constantly outperform it***
>
> As we said in the last question, our method relies on a good discriminator, which needs the map information. In our experiments, we obtain improvements in cases that maps are offered. The experiments on UCY where we are not always outperforming the baseline are more like a variant of our method since the maps are not accessible. To verify this statement, we run a new ablation experiment. We remove the map input to the discriminator of our model on nuScenes dataset and keep the remaining part unchanged (the map input to the generator is kept). The results are shown in Table 7 of the revised version. Without the map input to the discriminator, the performance drops by about 10% as we expected because we cannot learn a good discriminator. We revised our abstract section for a more accurate understanding.
>
> ***Q5 It would be interesting to see several runs of the same experiments also reporting the standard deviation of the obtained metrics. At least for the comparison with Trajectron++.***
>
> Thanks for your suggestion! We updated Tab.1 to report the mean and standard deviation average over 3 runs. Our method shows solid improvement over Trajectron++ when the prediction horizon increases.
>
>
> [1] Tang, Charlie, and Russ R. Salakhutdinov. "Multiple futures prediction." Advances in Neural Information Processing Systems. 2019.
>
> [2] Salzmann, Tim, et al. "Trajectron++: Multi-agent generative trajectory forecasting with heterogeneous data for control." European Conference on Computer Vision. 2020.

---

> ### Author Response · Authors · 2020-11-21
> **Part 2 Our Results' Significance**
>
> ***Q6 The proposed method’s performance is still somewhat close to Trajectron++’s***
>
> The FDE improvement over Trajectron++ is about 8% in Table 1. The significance of our method becomes more apparent on a longer prediction horizon as we show in Table 3. In the 6-second prediction case, we reduce FDE-Full from 6.70 to 4.93. This is 1.76 meters, and the improvement is about 27%. Besides, the generated trajectories from our method have a clear higher quality compared to our baseline measured by human study shown in Appendix D. We obtain 39.6% more votes than our baseline. In addition, our method reduces the road boundary violation rate by a large margin from 19% to 9% reported in Table 3. This is also a significant improvement.

---

### Author Response · Authors · 2020-11-19
**HalentNet Paper Revised Version**

We thank reviewers for their valuable and insightful feedback.  We are encouraged that they found our method interesting (R3, R4), and our writing clear (R2, R3), our ablation studies systematic, method solid,  and easy to understand (R4), has a lot of potentials (R2),   human studies as interesting (R3). We are replying to the reviewers' comments  individually and  are incorporating all the feedback.

**Updated Paper**   We also revised our paper to incorporate reviewers' feedback. The changed text is marked in blue color for an easy and quick check. We list here the key updates that we applied to improve the paper.

1) We added new ablation studies to justify the reason for our particular design of $L_h$ and why other choices are less effective in Table 8 of Appendix C.  We also added another ablation on the importance of map information to the discriminator shown in Table 7 of Appendix C.

2) To improve clarity,  we added an algorithm of our method showing the detailed steps in Appendix G, including how we generate the classified intent and hallucinated intent and how we use them to generate trajectories.  For reference, we also attach the training loop code in Appendix H.

3) We enriched the discussion of the interplay between the classified intent and hallucinated intent in the ablation subsection for a better explanation (page 8-9).

4) We revised our introduction section to improve clarity.  We also updated the related work section to incorporate more references suggested by the reviewers.

---

### Decision · Program_Chairs · 2021-01-07
**Final Decision**

**Decision:**

Accept (Poster)

**Comment:**

The paper presents a method for future trajectory generation. The main contribution is in proposing a technique for data augmentation in the latent space which encourages prediction of trajectories that are both plausible, but also different from the training set. The results clearly show superior performance on standard benchmarks. The evaluation is thorough and ablations show that the proposed innovation matters.

R2, R3, R4 recommend that the paper be accepted with scores 6, 8, and 6 respectively. R1 recommends the paper be rejected with a score of 5. The main concern of reviewers are:

R1: " In summary, the paper suffers from lack of a clear justification of the proposed contributions, unfair evaluations, and questionable significance of the results." The authors addressed this concern in their rebuttal.

R2: "Some other points remain still open such as the limited focus on Trajectron in evaluations." Since trajectron is a recent SOTA, I think this is not a big concern. Authors compare against other baseline methods too.

R4: Comparison to Mercat, Jean, et al., ICRA 2020 is missing. The authors mention that their code is unavailable and therefore cannot compare.

R4: "underlying reasons for the success of different components (classification of latent intent and hallucinative latent intent) are hard to explain". I agree with this and this is also my major concern which I detail below.

The paper proposes to find diverse trajectories by generating two latent vectors: z, z'. The first h time steps are generated by latent vector z and the remainder using z'. The generated trajectory is evaluated by a discriminator that ensures plausibility. The latent vectors are chosen to be discrete and a classifier is trained to recognize z from ground truth trajectories. To encourage diverse trajectories, authors use a loss that encourages mis-classification of the latent variable inferred from the generated trajectory. Since the generated trajectory cannot be classified well, it is assumed to be different from the training set.

This formulation is rather adhoc. If the trajectory is indeed different from the training distribution, then it will also fool the discriminator. If it doesnot, then it's not very different. The mis-classification, is akin to encouraging high entropy in the z space inferred from predicted trajectories. With this view, it is possible that there is no need to generate two latent vectors z, z', but simply generate one and use the entropy penalty. I would love to see this experiment and see the authors demystify their method. It would also lead to significant changes in writing. Even now, writing needs improvement. Due to the proposed method being a adhoc trick, that is not well justified, I would normally not recommend acceptance. However, the empirical results are strong, tilting the recommendation to acceptance.